

# Impact of two field preservation methods on genotyping success of feces

Valentina Valencia-Montoya[*], Isabel Salado[*], Ines Sanchez-Donoso, Alberto Fernández-Gil, Carles Vilà and Jennifer A. Leonard

Estación Biológica de Doñana, Consejo Superior de Investigaciones Científicas, Seville, Spain
[*] These authors contributed equally to this work.

## ABSTRACT

Non-invasive samples, such as feces, remain an important source of DNA for genetic analyses in molecular ecology and conservation genetics, especially when working with elusive or endangered species. However, as labs transition to higher throughput and genomic-based technologies, many protocols that have been used for decades are becoming obsolete. New approaches have been developed for high-quality samples, now low-quality samples require further technical advances. Fecal samples obtained for non-invasive wildlife studies are typically of very low quality and sampling methods need to be optimized to reduce work and costs per sample. Preservation methods in the field affect the workload in the lab required to obtain genetic data, as well as the final genotype quality. Liquid preservation methods, such as nucleic acid preservation (NAP) buffer and ethanol, have been used during sampling to maintain DNA quality at room temperature until samples can reach the lab. NAP buffer is a non-hazardous, non-flammable solution (easy to send through post), and avoids having to dry the feces before DNA extraction (saving time and increasing safety). Here we compare two different liquid preservation methods (NAP buffer and 96% ethanol) for microsatellite genotyping by next generation sequencing of wolf fecal samples collected in the field and shipped at ambient temperature. Samples preserved in ethanol showed a higher rate of amplification and genotyping success than in NAP buffer, especially due to a higher rate of allelic dropout in NAP. Consequently, the number of replicates required to achieve high quality genotypes was slightly higher for fecal samples preserved in NAP buffer than for those preserved in ethanol. These results are important for the planning and optimization of projects that involve microsatellite genotyping from feces using high throughput technologies.

Corresponding author
Jennifer A. Leonard,
jleonard@ebd.csic.es

## INTRODUCTION

An important source of data for many molecular ecology and conservation genetic studies is feces collected from animals in the wild. Feces collection enables the genetic study of elusive and endangered animals without disturbing them, through a non-invasive sampling (*Taberlet et al., 1999*). The analysis of fecal DNA can document the presence of species or the identification of individuals (*e.g., Kohn et al., 1999*; *Mumma et al., 2015*; *Stenglein et al., 2010a*; *Stenglein et al., 2010b*) and yields insights into spatial ecology and social behavior

(*Forcina et al., 2019*; *Goldberg et al., 2020*; *Ibáñez et al., 2016*). Although this particular type of sample is a valuable window into the ecology, behavior and population biology of wild species, it also has some disadvantages. The DNA extracted from feces is generally degraded, has low concentration, and is mixed with many unknown inhibitors and DNA from other species (*Deagle, Eveson & Jarman, 2006*; *Kohn & Wayne, 1997*). These features create logistical hurdles and complicate the high throughput processing of DNA from feces.

Fecal DNA is extensively used to amplify fragments of mitochondrial DNA for host identification (barcoding) and genotyping with nuclear microsatellites (*e.g.*, *Galan, Pagès & Cosson, 2012*; *Gil-Sánchez et al., 2020*; *Muñoz Fuentes et al., 2009*). These techniques have been heavily dependent on gel electrophoresis for separating fragments either for Sanger sequencing or genotyping of microsatellite loci by length polymorphism. The equipment necessary to perform these analyses can not process very many samples at a time (generally up to 96) and are becoming obsolete. However, novel approaches have been developed in recent years that involve the use of massive sequencing approaches that can revitalize the use of these genetic markers by scaling up and solving some of the main problems that microsatellite typing had as compared to single-nucleotide polymorphism (SNP) typing, such as low throughput and automation, as well as lack of transferability across laboratories, or even between projects within the same laboratory (*De Barba et al., 2017*; *Guichoux et al., 2011*; *Zhan et al., 2017*). The optimal use of resources requires every step of the process, from experimental set-up and sample collection in the field (*Sarabia et al., 2020*) to genotyping (*Salado et al., 2021*), to be revised and optimized to be compatible with genomics technology. Here we focus on the impact of field preservation of samples on genotyping success.

The most common field preservation method is to put the fecal sample (whole or fragments) in 70–100% ethanol to ship it to the lab at room temperature (*de Oliveira, Gonçalves & Galetti Jr, 2025*; *Panasci et al., 2011*). Dry preservation (for example with silica) has also been shown to preserve DNA (*Frantzenm Ma et al., 1998*), but is less practical in humid environments. Working with ethanol has some disadvantages. First, it is volatile, so the concentration may change substantially if the tube is left open for a while or not completely sealed, resulting in a much lower ethanol concentration (worse for DNA preservation). If several samples are collected, the volume of ethanol can become a problem for shipping because it is flammable. Finally, once in the lab, samples preserved in ethanol need to be completely dried before processing because even a small amount of ethanol will denature enzymes, such as proteinase K. This drying process takes time and may increase the risk of contamination while tubes are left open in the lab. Lastly, whether the sample was kept dry since collection or dried in the lab, handling dry feces increases exposure to air-borne particles -including parasite eggs- for laboratory personnel (*Sánchez Thevenet et al., 2003*). A potential solution to these problems would be the use of nucleic acid preservation (NAP) buffer (*Camacho-Sanchez et al., 2013*). Similar to RNA*later*® (Qiagen), NAP buffer is a salt solution (see Methods for composition details) that stabilizes and protects DNA and RNA from degradation by permeating the tissue and inhibiting enzymatic activity. NAP buffer is non-hazardous and non-flammable, making it easy to ship. Also, samples preserved in NAP do not need to be dried before DNA extraction,

which shortens processing time, decreases the risk of contamination, and reduces the risk to the technician of inhaling disease vectors.

NAP buffer preserves DNA quality and quantity slightly better than 95% ethanol for high quality tissue samples, but high molecular weight DNA is preserved in both conditions (*Camacho-Sanchez et al., 2013*). However, samples such as feces, which contain a wide variety of contaminants, may provide other challenges when preserving DNA. Ethanol is the most common field preservation method for fecal samples (*Panasci et al., 2011*), but the impact of these two preservation methods on genotyping success of fecal samples has not been previously assessed using next-generation sequencing data. Here we evaluate the quality of genotypes from fecal samples preserved in ethanol and in NAP buffer. We hypothesize that both preservation methods will perform similarly and no difference in the quality of genotypes of fecal samples will be found.

## MATERIALS AND METHODS

### Materials

Twelve feces presumed to correspond to Iberian gray wolves (*Canis lupus*) were collected in northern Spain in July 2020 (Table S1). In the field, a 5-cm fragment (approximately 20 g, wet weight) was taken from each feces and submerged in 30 mL 96% ethanol in a 50 mL Falcon tube (0.67:1, sample:solution), and another fragment of the same size in a tube with the same amount of NAP buffer (Fig. 1). The NAP buffer consisted of 0.019 M Ethylenediaminetetra-acetic acid (EDTA) disodium salt dihydrate, 0.018 M sodium citrate trisodium salt dihydrate, and 3.8 M ammonium sulphate, and was adjusted to pH 5.2 with $H_2SO_4$; (see *Camacho-Sanchez et al., 2013* for the full protocol). All samples were kept at ambient temperature for three weeks before shipping to the lab. Feces were received at the end of July 2020 and stored at −80 °C until processing in the lab in February 2021.

### Lab methods

DNA was extracted from a sample of approximately 150 mg of the side of the scat (*Stenglein et al., 2010a*; *Stenglein et al., 2010b*) per preservation media and negative controls with a silica-based method (*Höss & Pääbo, 1993*) in an isolated, dedicated low-quality DNA extraction lab. Seven polymerase chain reaction (PCR) replicates of a multiplex of 32 loci were amplified from each extract, plus an extraction negative and a PCR negative (Fig. 1). The multiplex included autosomal microsatellite loci, mostly dinucleotide repeats, with alleles ranging in size from 52 to 228 bp (Table S2; *Breen et al., 2001*; *Francisco et al., 1996*; *Fredholm & Winterø, 1995*; *Jouquand et al., 2000*; *Mellersh et al., 1997*; *Ostrander, Sprague & Rine, 1993*; *Ostrander et al., 1995*). We selected the best-performing 32 autosomal microsatellite loci from a previous study that applied high throughput sequencing to a larger microsatellite panel (46 loci, *Salado et al., 2021*). Multiplex reactions included 1X Phusion U Green multiplex PCR Master Mix, 0.05 µM of each primers, 0.8 mg/mL of bovine serum albumin (BSA) and two µl of DNA extract in a total volume of 20 µl. Amplification program was 98 °C for 1 min, 10 cycles of 98 °C for 10 s, 67 °C −1 °C per cycle to 56 °C for 30 s, and 72 °C for 30 s, followed by 20 cycles of 98 °C for 10 s, 56 °C for 30 s and 72 °C for 30 s with a final extension at 72 °C for 10 min and 95 °C for 3 min.

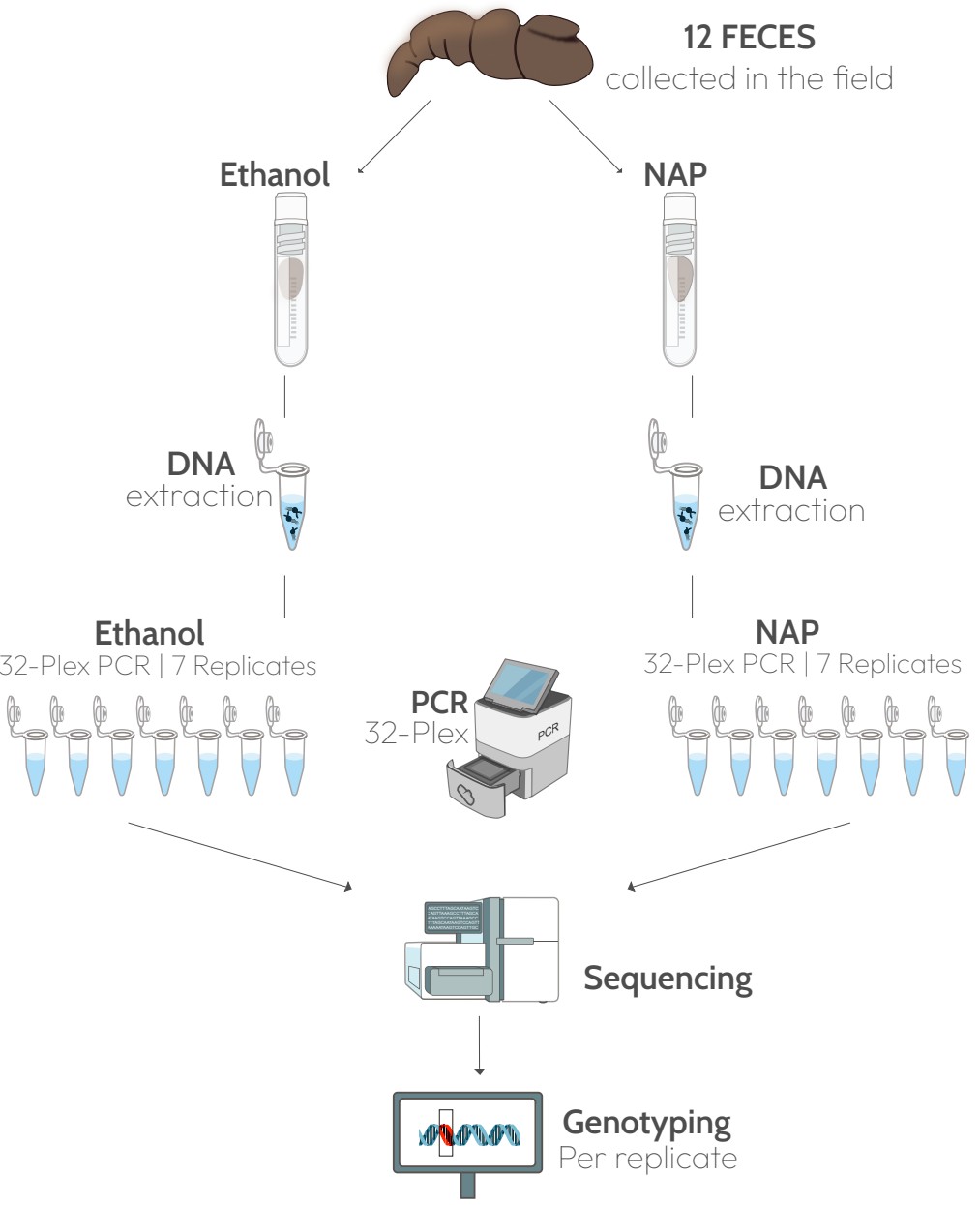

**Figure 1 Experimental design.** Twelve fecal samples were collected in the field and a fragment of each feces was preserved in ethanol 96% and another in NAP buffer. DNA was extracted from the two fragments and 32 microsatellite loci were amplified in a multiplex PCR. Each multiplex amplification was replicated seven times. Graphic materials are from public domain by NIH-Bioart: NIAID Visual & Medical Arts. 26/09/2024: Next Gen Sequencer (NIAID NIH BIOART Source, Bioart.niaid.nih.gov/bioart/386), Cryogenic Vial (88), Eppendorf Tube (143), qPCR Machine (426), DNA (124), Gene Mutation (170).

All PCR products were cleaned twice with 30 µl 1.5X SPRI beads (*Rohland & Reich, 2012*). Positive reactions were individually double-indexed for Illumina sequencing in PCR reactions with 1X Kapa Hifi HotStart ready Mix, 0.42 µM of each index, and six µL of

cleaned PCR product in a total volume of 12 μl, and cycling conditions of 95 °C for 3 min, 10 cycles of 98 °C for 20 s, 60 °C for 15 s, and 72 °C for 20 s with a final extension at 72 °C for 1 min. The indexed product was checked and quantified on an agarose gel using ImageLab v.6.1 (Bio-Rad). Finally, all products were pooled equimolarly for paired-end sequencing on a MiSeq (Illumina) with 300 cycles in the genomics core facility at John Hopkins University (Baltimore, MD, USA).

## Genotyping and error rates

Each PCR replicate and locus was independently genotyped in Megasat (v1.0, *Zhan et al., 2017*) using the forward sequence as input, and default settings except for a minimum read depth of 20 reads required for genotype calling. We considered three different error categories when a genotype was not called (according to Megasat): 'Not amplified' when a locus doesn't occur in a sample (labeled as 'X X' by Megasat), 'Unscored' when there are three possible real alleles and thus genotype is difficult to be determined by the program and, 'Insufficient Coverage' when depth of alleles is too low to score a genotype (labeled as '0 0' by Megasat, < 20 reads, see above) (Table S3). We established the *reference genotype* as the most likely genotype based on the consensus of all 14 replicates across the two treatments, following the criteria from *Salado et al. (2021)*. In brief, heterozygotes were called if each allele was present in at least two replicates, and homozygotes if the same allele was present in three or more replicates and no other allele was present more than once. Genotypes that did not meet these quality thresholds were labeled as 'ambiguous' and no genotype was called (Table S4).

Errors are especially common when genotyping microsatellite loci from fecal DNA (*Bonin et al., 2004*; *Pompanon et al., 2005*; *Taberlet et al., 1999*). We considered a genotyping error as any allelic difference between the reference genotype and the genotype obtained for each PCR reaction. We considered two main types of errors: allelic dropout as the failure to amplify one allele from a heterozygote (*Taberlet et al., 1996*), and false allele implies the appearance of an allele that is not present in the reference genotype (*Pompanon et al., 2005*).

To compare the two preservation media, we defined the following variables. Amplification success (AS) was the proportion of PCR replicates yielding reads assigned to a locus by Megasat, averaged across loci. Genotyping success (GS) was the proportion of positive PCR replicates at each locus with a correct genotype, that is, a coinciding genotype with the reference genotype, averaged across loci. The rate of allelic dropout (ADO) was the number of heterozygous genotypes where only one allele was detected divided by the total number of heterozygous reference genotypes. The rate of false alleles (FA) was the proportion of genotypes containing a false allele. We calculated ADO and FA using equations (2) and (4) from *Broquet & Petit (2004)*.

## Statistical analyses

We statistically compared the effect of the two preservation media over AS and GS by fitting two generalized linear mixed models (GLMM) under a negative binomial family (nbinom2) with the function glmmTMB() from the package glmmTMB (*Brooks et al.,*

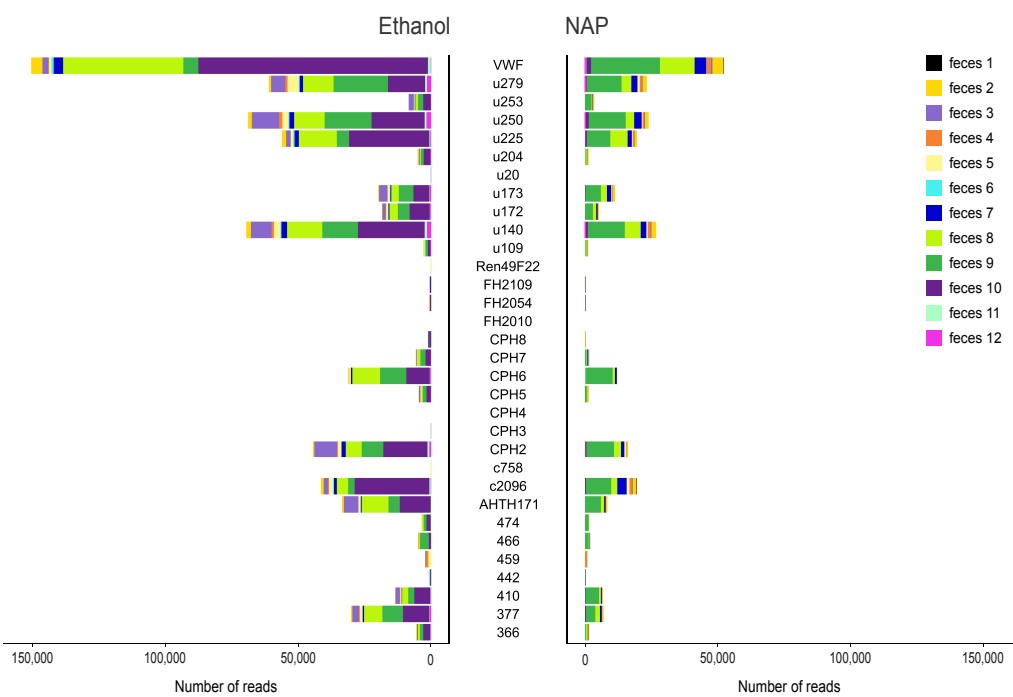

**Figure 2  Variance in amplification success across 32 microsatellite loci.** Loci genotyped in twelve gray wolf fecal samples preserved in ethanol 96% (left) and NAP buffer (right). The horizontal axis represents the total number of reads per locus obtained for the different feces (highlighted in different colors).

*2017*). We used PCR replicate for each locus and feces as sample unit in both models, and the two levels for the response factors AS and GS were failure (0) or success (1) (Table S3). As we observed differences in the total number of reads assigned to a locus between the two preservation media (see Results, Fig. 2), we added coverage (*i.e.,* number of reads per PCR replicate/locus/feces) as a fixed factor in the models (Table S3). We scaled coverage, by using the scale() function, which standardized the variable by subtracting the mean and dividing by the standard deviation. We also controlled the effect of the differences among feces and locus by including them and their interaction as random factors in the models. Normality of residuals was checked by visual exam of scatterplots. Significance was evaluated with a Chi-square test using the Anova() function available in the R car package (*Fox & Weisberg, 2018*). Statistical analyses were performed with R version 4.4.2 (*R Core Team, 2024*) by using RStudio version 2024.12.0 (*Posit team, 2024*).

## Estimation of number of replicates needed for genotyping

To have an assessment of the potential impact of the two preservation media on the cost and reliability of a feces genotyping project, we assessed how the preservation media affected the number of replicates needed to genotype individual feces. As a simplification of the complexities associated with genotyping, we focused on homozygous loci and assumed that we needed at least three identical correct genotype calls in separate PCR replicates to assign a genotype to a locus. We also assumed that all loci had the same probability of

**Table 1  Amplification and genotyping success, and error rates by preservation method.**

| Preservation method | AS | GS | ADO | FA |
|---|---|---|---|---|
| Ethanol | $0.62 \pm 0.02$ | $0.87 \pm 0.03$ | $0.17 \pm 0.03$ | $0.02 \pm 0.00$ |
| NAP | $0.52 \pm 0.02$ | $0.77 \pm 0.04$ | $0.27 \pm 0.03$ | $0.05 \pm 0.01$ |

**Notes.**
AS, amplification success, proportion of PCR replicates yielding reads assigned to a locus by Megasat, averaged across loci; GS, genotyping success, proportion of positive PCR replicates at each locus with a genotype coinciding with the reference genotype, averaged across loci; ADO, rate of allelic dropout, calculated as the number of heterozygous genotypes where only one allele was detected, divided by the total heterozygotes in the reference; FA, rate of false allele, proportion of genotypes containing a false allele.
Reported values represent the mean $\pm$ standard error (SE) per locus.

offering a correct genotype. We used a binomial distribution to calculate the probability of correctly genotyping one homozygous locus after a given number of replicates taking into account AS and GS for each preservation media (see Results, Fig. S1, Table 1). We then used this probability to estimate the probability of obtaining a multilocus genotype for a subset of 15, 20 or 25 loci or more using also a binomial distribution.

## RESULTS

### Data

Sequencing produced a total of 8,970,484 raw reads (mean $\pm$ standard error SE per multiplex PCR product: $28,209 \pm 1,662$), although a total of 28 PCR replicates were not successfully sequenced (*i.e.,* those that yielded 0 or a few tens or hundreds of reads per PCR replicate; 12 for NAP, 16 for Ethanol; Table S5). The total number of raw reads from ethanol preserved samples was 4,987,792 (mean $\pm$ SE per multiplex PCR product: $30,788 \pm 2,740$), and 3,982,692 for NAP preserved samples ($25,530 \pm 1,824$). In general, the number of reads assigned to a locus per PCR varied substantially across markers (mean $\pm$ SE: from $0 \pm 0$ for locus FH2010 to $1,274 \pm 311$ for locus VWF; Fig. 2). Overall, we observed a lower total number of reads per locus in NAP than in ethanol. There was also a high variance in the number of reads across samples. For example, some samples (*e.g.,* feces #3, 8 and 10) preserved in ethanol provided a higher number of reads than with NAP, while others did not work at all in either of the two preservation media (*e.g.,* feces #1) (Fig. 2).

### Amplification and genotyping success

We obtained 184 genotypes (109 heterozygotes, 75 homozygotes) out of 384 total possible genotypes (12 feces $\times$ 32 loci) to be used as reference (Table S4). From those without a genotype, 48 were considered 'ambiguous' following our genotyping criteria (see Methods). For downstream analyses, we only considered the 184 reference genotypes. All fecal samples yielded more than ten reference genotypes (out of 32), except for three (feces #1, 6 and 11; Fig. S2). GLMM analyses showed that samples in ethanol had a higher Amplification success (AS $= 0.62$, $Chisq_1 = 16.88$, $p < 0.001$), and also a higher Genotyping success (GS $= 0.87$, $Chisq_1 = 25.28$, $p < 0.001$) than those preserved in NAP (AS $= 0.52$; GS $= 0.77$) due primarily to a higher rate of allelic dropout in the NAP preserved samples (ADO NAP $= 0.27$; ADO Ethanol $= 0.17$) (Table 1; Fig. 3). Coverage (*i.e.,* number of reads per PCR replicate/locus/feces) showed a significant effect only over amplification success (AS,

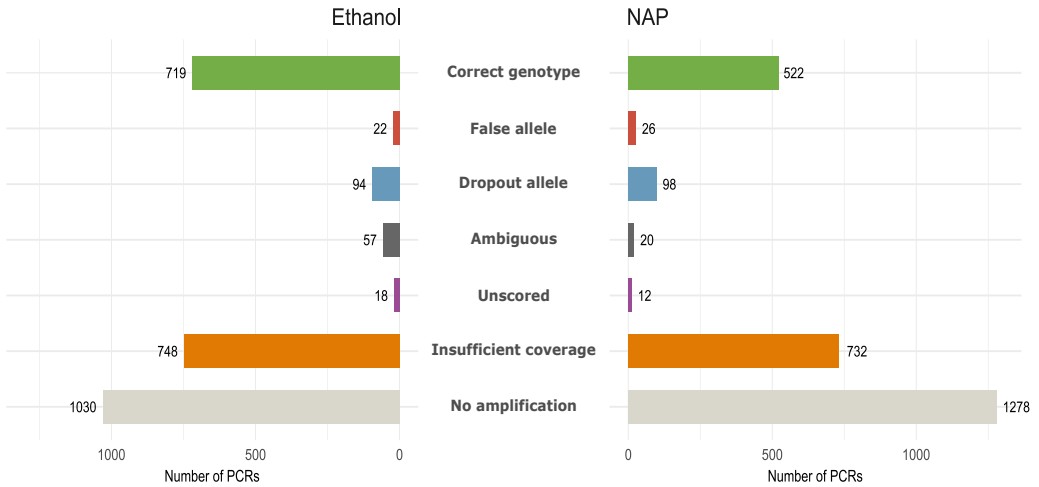

**Figure 3** **Amplification and genotyping success and genotyping errors by preservation media.** Number of PCR reactions and locus categorized according to genotyping error or amplification failure (see Methods) in the same 12 fecal samples preserved in either ethanol or NAP buffer (total number of PCRs per preservation method: 12 feces ×7 replicates ×32 loci= 2,688).

Chisq$_1$ = 14.44, $p < 0.001$), meaning that a higher coverage leads to obtaining more PCR replicates with a genotype, but not necessarily with a genotype that is correct.

## Number of replicates

Both the amplification and the genotyping success were higher for samples preserved in ethanol. To assess the potential impact of these differences on the cost and reliability of a feces genotyping project, we calculated how they affected the number of replicates needed to genotype individual feces for different numbers of loci. Results showed that the number of replicates necessary to have high quality genotypes was lower for samples preserved in ethanol. For example, for samples preserved in ethanol, seven replicates would be needed for obtaining a correct genotype with a probability higher than 95% for 20 of the 32 microsatellites, while samples preserved in NAP would require nine replicates (Fig. 4).

## DISCUSSION

The equipment, materials and software necessary for "traditional" genotyping of microsatellite loci are very quickly becoming obsolete as the industry moves towards genomics technologies. However, microsatellite loci can still be very informative in some studies (*Cueva et al., 2024*; *Cullingham et al., 2023*; *Karamanlidis et al., 2021*; *Riquet et al., 2021*), and adapting microsatellite typing protocols to next generation sequencing technologies can greatly facilitate their use (*De Barba et al., 2017*; *Lanner et al., 2021*). Experiments need to be carefully planned because one characteristic of "next generation" technologies is that they generate huge amounts of data per run, so larger units of work need to be planned at a time. The careful planning of each step can have a large impact on the quality of the final dataset (*Brandariz-Fontes et al., 2015*).

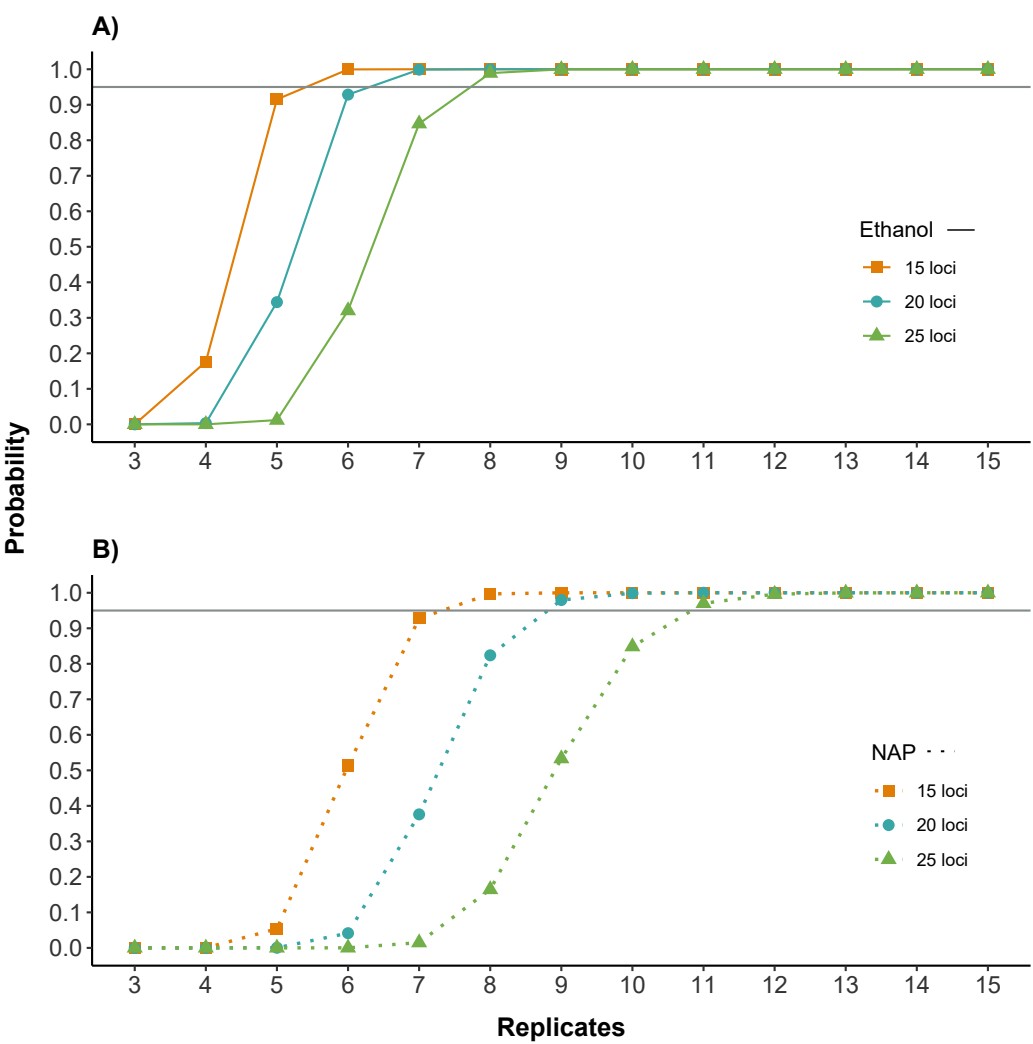

**Figure 4 Number of replicates needed to obtain reliable genotypes for different numbers of markers using (A) ethanol (B) NAP buffer as DNA preservation media.** Probability of obtaining a correct genotype for at least 15, 20 or 25 loci out of the panel of 32 microsatellites for both preservation media. Calculations are based on the probability of obtaining a correct genotype for a homozygous locus by obtaining at least three coinciding genotype calls (note that this is the criterion used to confirm genotypes; see Methods) in a given number of PCR replicates (see also Fig. S1 for single locus). Horizontal line marks probability of 0.95. The number of replicates required to obtain a correct multilocus genotype is higher for NAP preserved samples.

Various steps in the process of genotyping feces, from field collection (*Sarabia et al., 2020*; *Scholz et al., 2024*; *Van Cise et al., 2024*), to lab analysis (*Sarabia et al., 2020*), to bioinformatics (*De Barba et al., 2017*; *Salado et al., 2021*), have previously been examined to improve compatibility with high-throughput workflows. Another important step that could be optimized is preservation of samples in the field for transport to the lab. A review of the literature in Web of Science (keyword strings TS= ((noninvasive OR non-invasive OR "non invasive" OR feces OR fecal OR faecal OR faeces OR scat OR

saliva OR hair OR environmental DNA OR eDNA) AND (genetic* OR molecular* OR DNA) AND (species OR wildlife) AND (microsatellite OR STR OR short-tandem repeats) AND (high throughput OR HTS OR next-generation OR next-generation OR NGS) AND (amplification success OR genotyping success))) returned 26 papers, of which only four both reported how fecal samples were preserved in the field and then were genotyped with next generation sequencing tools (*Barbian et al., 2018*; *De Barba et al., 2017*; *Salado et al., 2021*; *Sarabia et al., 2020*). These papers employed different protocols on different sample types from different species in a variety of habitats and yielded highly variable results (Table S6). The clearest result is that few papers describe their collection methods clearly, but these data would be an important addition to the literature so that future meta-analyses could be performed to better understand DNA preservation and degradation under practically important conditions.

In this study, we hypothesized that both preservation methods should preserve DNA similarly well, however we found that there was a difference between amplification and genotyping success between fecal samples preserved in ethanol and those preserved in NAP buffer. Those preserved in ethanol had a lower rate of allelic dropout and would require fewer replicates to obtain reliable genotypes at multilocus genotypes. However, the number of replicates will also depend on the research question, the specific panel of loci, and the sample type. Our estimated number of replicates are panel-specific. Since the difference between the two preservation methods in the number of replicates required to reach confidence thresholds per locus is not large, it could be relevant to also consider other benefits or disadvantages of the preservation methods when designing an experiment. Other potentially relevant characteristics include safety and processing time. NAP buffer is safe to ship because it is not flammable, and also increases safety of the lab personnel by enabling the feces to be extracted directly without drying. This can also reduce handling time, and thus overall lab processing time.

Here we have focused solely on preservation for amplification of DNA markers, but similarly preserved samples could be used for a variety of other molecular analyses as well, and ethanol and NAP could be better or worse for those other analyses. RNA, for example, does not preserve in ethanol but should preserve in NAP (*Camacho-Sanchez et al., 2013*). Faecal and environmental microbiota also preserve well in NAP (*Menke et al., 2017*; *Ward et al., 2023*). Different preservation methods also impact stable isotopes differently (*Barrow, Bjorndal & Reich, 2008*). Since the same feces may be used for a variety of different analyses (*Gil-Sánchez et al., 2020*), the appropriateness of preservation for the other analyses should also be taken into account.

Additionally, our results showed that there was a large variance in the amplification success for the different microsatellite loci (Fig. 2). All of the loci included here have previously been used by our group with "traditional" microsatellite analyses involving size separation in polyacrylamide gel (*Hailer & Leonard, 2008*; *Koblmüller et al., 2009*; *Musiani et al., 2007*). Some loci that worked well with traditional methods did not work well or at all with next generation sequencing, as *Salado et al. (2021)* also found with a larger multiplex microsatellite panel (46 loci), that included all the microsatellite loci used in this study. For this panel, *Salado et al. (2021)* recommended a coverage higher than 100

reads per PCR replicate/locus/feces using the same bioinformatic genotyping method as this study. *De Barba et al. (2017)* designed a new 13-microsatellite panel specifically for the analysis of degraded DNA samples with the high-throughput sequencing approach in brown bear (*Ursus arcto*s) and obtained an improvement of genotyping success (84%) and cost reduction compared to capillary electrophoresis. Determining which characters make loci more appropriate for high throughput sequencing will be important in the planning and optimization of projects including microsatellite genotyping in the future.

We observed differences in amplification and genotyping success across fecal samples (Fig. 2, Fig. S2). An interaction between preservation methods and diet content has also been reported to impact both amplification and genotyping success in fecal DNA using traditional genotyping, specifically for the probability of false alleles (*Panasci et al., 2011*). Ethanol preservation has been suggested for scats of obligate carnivores and of facultative carnivores with a diet consisting primarily of mammals, while salt-solution-based buffers (*e.g.*, dimethyl sulfoxide saline solution, DET buffer- composed by 20% DMSO, 0.25 M EDTA, 100 mM Tris, pH 7.5 and NaCl to saturation; *Seutin, White & Boag, 1991*) are suggested for preservation for scats of animals with a diet consisting of plant-derived foods (*Panasci et al., 2011*). Our wolf feces contained mainly mammals, although some included plant material, as wolves sometimes swallowed them accidentally with other food or for curative purposes (*Pezzo, Parigi & Fico, 2003*). However, our sample size is too limited ($n = 12$ feces) to find a trend between fecal content and preservation methods. Future studies could evaluate the interaction between preservation methods and fecal content and how this impacts the amplification and genotyping success in feces using genotyping based on high throughput data.

## CONCLUSIONS

Ethanol is better at preserving feces for genotyping microsatellites with Illumina sequencing, but the difference between preservation methods does not result in dramatic differences in the number of replicates required to generate high quality genotypes. Thus, other considerations such as logistics and safety should be taken into account when selecting a field preservation solution. The difference in success between loci and samples is much more important than the difference in field preservation, so careful sample and locus selection will be an effective way to increase the quality of genotypes for the same effort.

## ACKNOWLEDGEMENTS

We are very grateful to Anna Cornellas and Marta Portolà for their assistance in laboratory work. Logistical support was provided by Laboratorio de Ecología Molecular (LEM-EBD). We acknowledge three reviewers for their constructive feedback that improved the quality of the final manuscript.

### Funding

This research was funded by Junta de Andalucía ("Proyectos de generación de conocimiento", P18-FR-5099) to Jennifer A. Leonard and Carles Vilà. Isabel Salado received a PhD fellowship, "Ayudas para la formación de profesorado Universitario (FPU)" from the Spanish Ministry of Universities (FPU17/02584) and a research contract funded by "Asociación Apadrina La Ciencia-Ford Spain", with the support of Ford Motor Company Fund. This research was also partially funded by the WOLFNESS project within the Biodiversa+ program, the European Biodiversity Partnership under the 2021–2022 BiodivProtect joint call for research proposals, co-funded by the European Commission (GA N101052342) and the Spanish Ministry of Science and Innovation (PCI2022-135098-2). The CSIC Open Access Publication Support Initiative through its Unit of Information Resources for Research (URICI) supported the publication fee. There was no additional external funding received for this study. The funders had no role in study design, data collection and analysis, decision to publish, or preparation of the manuscript.

### Grant Disclosures

The following grant information was disclosed by the authors:
Junta de Andalucía ("Proyectos de generación de conocimiento"): P18-FR-5099.
The Spanish Ministry of Universities: FPU17/02584.
"Asociación Apadrina La Ciencia-Ford Spain", with the support of Ford Motor Company Fund.
The WOLFNESS project within the Biodiversa+ program, the European Biodiversity Partnership under the 2021–2022 BiodivProtect joint call for research proposals.
The European Commission: GA N101052342.
The Spanish Ministry of Science and Innovation: PCI2022-135098-2.
The CSIC Open Access Publication Support Initiative through its Unit of Information Resources for Research (URICI).

### Competing Interests

The authors declare there are no competing interests.

### Author Contributions

- Valentina Valencia-Montoya analyzed the data, prepared figures and/or tables, authored or reviewed drafts of the article, and approved the final draft.
- Isabel Salado performed the experiments, analyzed the data, prepared figures and/or tables, authored or reviewed drafts of the article, and approved the final draft.
- Ines Sanchez-Donoso analyzed the data, prepared figures and/or tables, authored or reviewed drafts of the article, and approved the final draft.
- Alberto Fernández-Gil performed the experiments, authored or reviewed drafts of the article, and approved the final draft.
- Carles Vilà analyzed the data, authored or reviewed drafts of the article, and approved the final draft.

- Jennifer A. Leonard conceived and designed the experiments, authored or reviewed drafts of the article, and approved the final draft.

## DNA Deposition

The following information was supplied regarding the deposition of DNA sequences:

The raw sequence data is available at NCBI: PRJNA1235509.

## Data Availability

The sample metadata, microsatellite loci information, coverage, genotype information, and R script used for statistical analyses and plots are available in the Supplementary File.

## Supplemental Information

Supplemental information for this article can be found online at http://dx.doi.org/10.7717/peerj.20154#supplemental-information.

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
