# Peer review of "Impact of two field preservation methods on genotyping success of feces"

_PeerJ, doi:10.7717/peerj.20154_

## Round 0.1 · original submission · Major Revisions

Dear Dr. Leonard and co-authors,

Thank you for submitting your manuscript to PeerJ. Your study on the comparative effectiveness of NAP buffer and ethanol as preservation media for microsatellite genotyping of fecal samples via next-generation sequencing provides important insights for non-invasive genetic studies in wildlife. The manuscript is generally well written, addresses a relevant question, and is methodologically sound. However, after reviewing the comments of three expert reviewers, we request that you address the following points in a revised version of your manuscript. We believe that these revisions will substantially improve the clarity, reproducibility, and impact of your study.

Major Concerns to Address
1. Methods Clarification and Completeness

Composition of NAP buffer: While cited in the Introduction, please restate the reference or explicitly list the composition in the Methods section for ease of consultation (Reviewer 1).

Sample-to-solution ratio: Please specify the volume of ethanol and NAP used per sample and the approximate mass of fecal material submerged, to allow reproducibility (Reviewers 1 and 3).

Data availability: Clearly state where the raw sequencing data and genotypes are stored (e.g., GenBank SRA, Dryad, Figshare) or explain why they are not publicly available (Reviewer 1).

2. Sample Size Limitations

The small number of samples (12) is a major limitation for the generalizability of your conclusions. While we understand logistical constraints, you should explicitly acknowledge this in the Discussion and elaborate on how this affects the robustness and applicability of your conclusions (Reviewer 2).

3. Comparative Context

Include comparative data or at least a short synthesis (possibly in supplementary material) showing amplification/genotyping success and error rates from similar studies using ethanol or NAP in other species or with other genotyping technologies (Reviewer 2).

Consider citing and integrating the suggested recent reference (Kim et al., 2025) if relevant.

4. Multiplex Efficiency and Targeted-NGS

Clarify whether this 32-locus multiplex PCR has been used previously in the same configuration. If not, state this explicitly and discuss the implications for comparability (Reviewer 1).

Discuss the relatively low genotyping yield (184/384 = 47.9%) in light of existing targeted-NGS studies using microsatellites. Is this a methodological limitation or expected given the sample type? (Reviewer 1).

5. Minimum Coverage and GLMM Variables

Provide a recommended minimum coverage per locus or replicate to ensure genotype accuracy (Reviewer 1).

Clarify what measure of coverage was included as a fixed effect in your models (average per locus or replicate?), and how it relates to genotyping accuracy (Reviewer 1 and 3).

6. Hypothesis Statement

Explicitly include a hypothesis at the end of the Introduction, based on the literature review, regarding expected performance differences between preservation methods. Revisit this hypothesis in the Discussion (Reviewer 2).

Minor Revisions
Abstract: Spell out “NAP” as “Nucleic Acid Preservation” buffer at first use (Reviewer 3).

Clarify ambiguous phrases (e.g., what technology is becoming obsolete in L59–61; what are the main problems of traditional microsatellite genotyping in L70).

Improve precision in word choice (e.g., “important” → “substantial” in L85).

Report the number of failed PCR replicates (Reviewer 3).

Include statistical significance values (p-values) in Table 1 (Reviewer 1 and 3).

Address typographical issues (e.g., “all kept” → “all were kept”, “equimolar” → “equimolarly”) (Reviewer 3).

Conclusion
I consider the manuscript to be a valuable contribution but recommend major revisions before it can be considered for publication. I encourage you to respond to each reviewer comment point-by-point, indicating how you addressed the issue in the revised manuscript or providing a rationale if no change was made.

I look forward to receiving your revised submission.

Sincerely,
Rodrigo Nunes-da-Fonseca

**PeerJ Staff Note**: Please ensure that all review, editorial, and staff comments are addressed in a response letter and that any edits or clarifications mentioned in the letter are also inserted into the revised manuscript where appropriate.

Reviewer 1 ·

Basic reporting

The study presented in this manuscript presents valuable information on the usefulness of two preservation solutions in the process of microsatellite typing from wildlife faeces samples. The manuscript is well written and develops on the topic described in the title. The number of figures and tables is adequate, and the information presented adds to the interpretation of results. Some details are missing in the methods, and the discussion fails to address certain corollaries, as described below.

Experimental design

METHODS:
1. The composition of the NAP buffer is not given, although a reference is given in the introduction. This reference could be repeated in methods to clarify where to consult the composition of the buffer.

2. There is no indication on the sample-to-solution ratio, nor on the approximate (in g) amount of sample taken. Although 50 ml tubes are mentioned, the amount of solution contained is not mentioned.

3. It is not clear if the raw data is publicly available. No references are given on repositories used to store and share the data.

Validity of the findings

DISCUSSION and CONCLUSIONS from the study

4. Has the 32-multiplex PCR described here been attempted before? Can the authors provide a reference where these micros have been studied together before? This should provide a reference to compare the efficiency of this method with regard to a traditional capillary electrophoresis system. In L200 – “We obtained 184 genotypes out of 384 total possible 201 genotypes”. So, the efficiency of microsatellite genotyping by targeted-NGS is low (below 50%). Is this due to the expected degradation of the samples? Are there other studies using targeted-NGS for microsatellite targeting in wolves or other species? Given all the above, do the authors recommend this strategy for microsatellite typing in faeces from wildlife settlements? The discussion would benefit from the view of the authors on this as opposed to other current alternatives.

5. Given your results, what is the minimum coverage (per locus or average on the overall loci) needed or that you would recommend for correct genotype assignment?

Additional comments

MINOR ISSUES.

6. L164 – Can you clarify if the coverage used as fixed effect was the average coverage per locus and replicate, or the average coverage per replicate (average of all loci per replicate)?

7. L210-212 – Could you clarify if this last sentence? You refer to “coverage” as the number of reads per locus or the average across loci? If the former, it is not odd that the number of correct genotypes does not increase? So, the additional genotypes gained by increased coverage are all “wrong” genotypes?
8. Shouldn’t Table 1 include p-values assigned to these differences?

·

Basic reporting

Language (Professional and unambiguous) - Yes

Literature reference - lacks some recent references related to fecal genotyping success, for example, Kim et al, 2025: Strategic Sampling of Eurasian Otter Spraints for Genetic Research in South Korea: Enhancing PCR Success and Data Accuracy.

Professional article structure, figures, and tables. Raw data shared - Yes.

Self-contained with relevant results to hypotheses - make a hypothesis statement at the end of the introduction about why the experiment was designed and what results researchers expected based on the literature review. Also, a similar statement can be added in the discussion, too, comparing results with the hypothesis.

Experimental design

Original primary research within Aims and Scope of the journal - Yes

Research question well defined, relevant & meaningful. It is stated how research fills an identified knowledge gap - Yes.

Rigorous investigation performed to a high technical & ethical standard - Sample size is too low (If possible should be increased)

Methods described with sufficient detail & information to replicate - Yes

Validity of the findings

I have the following concerns -

(1) Small sample size

(2) Lack of comparative assessment of genotyping success obtained in the present study and other published studies. For example, it will be better if authors make a supplementary file by compiling amplification success and genotyping errors across different studies undertaken using ethanol and NAP buffer to have a comparative assessment with their findings.

Reviewer 3 ·

Basic reporting

The manuscript is well-written. There are a few minor additions/corrections that are required for clarity:

Abstract: Please spell out NAP buffer at first instance (Nucleic Acid Preservation buffer).

Lines 59-61: ‘These features and the use of a technology that is rapidly becoming obsolete, create logistical hurdles and complicate the high throughput processing of DNA from feces (De Barba et al.,2017).’ Which technology is rapidly becoming obsolete? This is defined in the next paragraph, which is awkward to read and comprehend.
Line 70: ‘scaling up and solving some of the main problems that microsatellite typing had compared to’. Please define the main problems, especially those that are relevant here.
Line 85: ‘important’ is not the correct word here. Maybe ‘substantial’?
Lines 89-90: What is the salt content of NAP buffer? How does it actually work?
Line 120: Please make it clear that you individually indexed the reactions for Illumina sequencing. Were they single or double-indexed?
Line 188: How many PCR replicates were not successfully sequenced?

Experimental design

-

Validity of the findings

Line 106: Approximately 150 mg of feces? How did you standardize yields between the two protocols, as you have a fundamentally paired design? Variance in feces mass is (presumably) a non-negligible effect. You did not include fecal mass as a variable in your GLMM. Did you quantify how much DNA went into each PCR reaction?

Lines 206-208: Are these differences statistically significant? This seems like you should test with paired t-tests (or similar). I assume that lines 208-210 showing the Chi2 results are the associated statistical tests. If so, it would be best to directly link the statistical test to the results, probably as a single line, e.g.:

‘Samples in ethanol showed a higher Amplification success (AS = 0.62; Chisq1=16.88, p < 0.001), and also a higher Genotyping success (GS = 0.87; Chisq1= 25.28, p < 0.001) than NAP (AS = 0.52; GS = 0.77) due primarily to a higher rate of allelic dropout in the NAP preserved samples (ADO NAP = 0.27; ADO Ethanol = 0.17)(Table 1; Fig. 3).’

Lines 216-218: You infer a greater number of needed replicates for NAP in your example than you genotyped in your experiment. Your methods are based on a very simple binomial distribution model on only 12 feces from a single context using one specific multiplex panel. You should note that your model infers 9 replicates and that real data may need more or fewer replicates based on sample preservation, marker length, etc.

Additional comments

Some very minor typographical errors:

Line 101: ‘all kept’ --> ‘all were kept’
Line 125: ‘equimolar’ --> ‘equimolarly’

---

## Round 0.2 · accepted · Accept

Congratulations on the acceptance of the manuscript.

Reviewer 1 ·

Basic reporting

The manuscript has gained clarity with the corrections and additions in this new version.

Experimental design

Despite the low sample number, the experimental design is sound and well explained in the text.

Validity of the findings

The authors do not overconclude from their findings. Despite the low genotyping call of the method, they show the goodness and limitations of each preservation media.

Additional comments

Two very small typos:
- Table S4, 3 homozygous genotypes are not coloured
- L310 – Write EtOH in full, as in the rest of the text.

Reviewer 3 ·

Basic reporting

no comment

Experimental design

no comment

Validity of the findings

no comment

Additional comments

The authors have sufficiently addressed mine and the other reviewers' comments.